# How the Direction of Screws Affects the Primary Stability of a Posterior Malleolus Osteosynthesis under Torsional Loading: A Biomechanical Study

Felix Christian Kohler [1,*], Philipp Schenk [1,2], Paul Koehler [1], Britt Wildemann [1], Gunther Olaf Hofmann [1,3], Steffen Derlien [4], Uta Biedermann [5], Isabel Graul [1] and Jakob Hallbauer [1]

1 Department of Trauma, Hand and Reconstructive Surgery, Jena University Hospital, Friedrich Schiller University Jena, 07747 Jena, Germany; philipp.schenk@bergmannstrost.de (P.S.); paul.koehler@uni-jena.de (P.K.); britt.wildemann@med.uni-jena.de (B.W.); gunther.hofmann@med.uni-jena.de (G.O.H.); isabel.graul@med.uni-jena.de (I.G.); jakob.hallbauer@med.uni-jena.de (J.H.)

2 BG Klinikum Bergmannstrost, Research Executive Department, Merseburger Straße 165, 06112 Halle, Germany

3 BG Klinikum Bergmannstrost, Department of Trauma and Reconstructive Surgery, Merseburger Straße 165, 06112 Halle, Germany

4 Institute of Physiotherapy, Jena University Hospital, Friedrich Schiller University Jena, 07747 Jena, Germany; steffen.derlien@med.uni-jena.de

5 Institute of Anatomy I, Jena University Hospital, Friedrich Schiller University Jena, 07743 Jena, Germany; uta.biedermann@med.uni-jena.de

* Correspondence: felix.kohler@med.uni-jena.de

**Abstract:** Insufficient fixation of a posterior malleolus fracture (PM) can lead to posttraumatic complications such as osteoarthritis and chronic pain. The purpose of this biomechanical study was to test the hypothesis of whether the direction of PM screw fixation has an impact on the primary stability of osteosynthesis of a PM under torsional loading. PM fractures of 7 pairs human cadaveric lower leg specimens were stabilized with posterior to anterior (p.a.) or anterior to posterior (a.p.) screw fixation. Stability of the osteosynthesis was biomechanically tested using cyclic external torsional loading levels, in 2 Nm steps from 2 Nm up to 12 Nm, under constant monitoring with 3D ultrasonic marker (Zebris). The primary stability does not differ between both stabilizations ($p = 0.378$) with a medium effect size ($\eta^2_p = 0.065$). The movement of the PM tends to be marginally greater for the osteosynthesis with a.p. screws than with p.a. screws. Whether a.p. screws or the alternative p.a. screw fixation is performed does not seem to have an influence on the primary stability of the osteosynthesis of the PM fixation under torsional loading. Although osteosynthesis from posterior seems to be more stable, the biomechanical results in the torsional test show quite equivalent stabilities. If there is no significant dislocation of the PM, a.p. screw fixation could be a minimally invasive but stable surgical strategy.

**Keywords:** posterior malleolus; Volkmann triangle; primary stability; ankle fractures; syndesmosis

## 1. Introduction

The dorsal aspect of the distal tibia, which is the bony fixation of the posterior inferior tibiofibular ligament (PITFL), is commonly referred to as the posterior malleolus (PM). Frequent injuries of the posterior syndesmosis complex occur with an intact PITFL and a fracture of the PM [1]. On the one hand, the bony avulsion of the PITFL leads to the loss of one of the main stabilizers of the syndesmosis; on the other hand, the fragment can affect the articular surface of the upper ankle joint. This leads to damage of the articular cartilage and most likely plays an important role in the pathogenesis of long-term complications such as chronic pain and post-traumatic osteoarthritis [1,2].

It is well accepted that PM should be fixed if the fragment includes at least 25% or 33% of the tibiotalar articular surface [1,3]. However, no data exists for decisions based on percent joint surface involvement [4]. In their systematic review, Bartoníček et al. made the following recommendations for the indication of surgical treatment of PM fractures [5]:

- The original size and congruence of the articular surface of the tibial pilon and, thus, the posterior stability of the ankle should be restored;
- The integrity of the posterior tibiofibular ligament and, thus, the stabilizing function of the upper ankle joint should be restored;
- The integrity of the fibular notch should be restored to facilitate the reduction of the distal fibula, especially in high Weber type C fibular fractures, including Maisonneuve fractures.

The reduction can be performed indirectly using ligamentotaxis and minimal invasive fixation from the anterior in the case of easily reducible fragments or directly from the posterior in case of insufficient reducibility [6]. There is no clear recommendation regarding which approach and which osteosynthesis (OS) technique should be preferred [7–9].

Bartoníček et al. referred the direct reduction and p.a. OS as biomechanically more stable than indirect reduction and a.p. screw fixation [5]. However, biomechanical results comparing a.p. and p.a. screw OS are not yet available. The purpose of the biomechanical study presented here was to clarify whether the direction of screw OS of a PM fragment has a relevant impact on the primary stability under torsional loading.

In contrast to Bartoníček et al. [5], it was hypothesized that there is no biomechanical difference in primary stability between screw osteosynthesis of the PM with respect to the direction of screw insertion (p.a. versus a.p.) for stabilization of a fracture of a large PM fragment (type IV according to Bartoníček and Rammelt [9]).

## 2. Materials and Methods

### 2.1. Specimen Selection

Fresh frozen lower legs from seven body donors from the anatomical institute of the University of Jena were used in the study (6 males and 1 female, age: 76 $\pm$ 8 years). There were no known ankle injuries or surgeries and no musculoskeletal diseases in the donors' documented history. Approval was granted by the Ethics Committee of the Jena University Hospital (approval No. 2020-1766-material). Bone density was determined by dual X-ray absorptiometry (DEXA; Lunar Prodigy DF+ 502015, GE Healthcare, Madison, WI, USA), to exclude differences in bone quality between specimens.

### 2.2. Preparation of the Specimens

The lower legs were thawed overnight, and tested the following day. A soft tissue window was prepared dorsally above the PM to allow the osteotomy. All other ligamentous structures as well as the entire foot remained intact. A Steinmann pin (5 mm; DePuy Synthes) was used to perform a subtalar hindfoot arthrodesis in a neutral position to prevent motion in the lower ankle joint as much as possible. The PM was osteotomized exactly 4 cm above the joint level using custom-made saw blocks in the coronary plane, resulting in a type IV fracture according to Bartoníček and Rammelt (involving 25–33% of the articular surface). The tibial plateau was fixed in portable frames and clamped in the testing machine (ZwickiLine Z1.0 materials testing machine from Zwick/Roell, Ulm, Germany).

The specimens were stabilized as follows (see Figure 1):

- Right lower legs: The fragments were reduced by ligamentotaxis and stabilized with two bicortical fully threaded screws (3.5 mm; DePuy Synthes) a.p. (Figure 1b);
- Left lower legs: PM was reduced and two bicortical fully threaded screws (3.5 mm; DePuy Synthes) were inserted p.a. (Figure 1a). The PM was anatomically reduced to ensure accurate reconstruction of the PITFL and restoration of the articular surface.

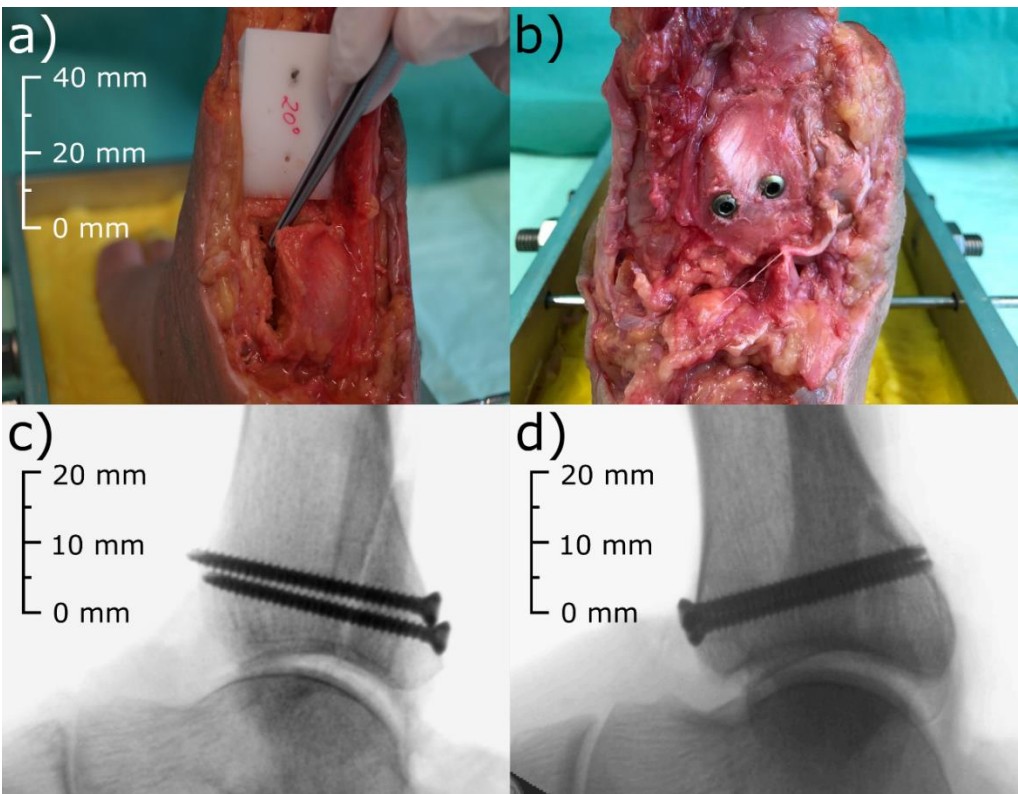

**Figure 1.** The figure shows in (**a**) the osteotomy of a fracture of the PM (type IV according to Bartoníček and Rammelt), in (**b**) the osteosynthesis by example using screws implanted from the posterior, in (**c**) the lateral X-ray image of the p.a. screw osteosynthesis and in (**d**) the a.p. screw osteosynthesis in the lateral X-ray.

During testing, constant monitoring of the OS of the PM was performed by a 3D ultrasound movement measurement system with an accuracy of 0.1 mm (CMS 70PV5, Zebris Medical GmbH). The ultrasonic surface markers were always positioned as follows (see Figure 2a,b):

- Marker 1 (M1): on the posterior malleolus (after osteosynthetic treatment);
- Marker 2 (M2): on the tibia 1 cm above the fracture level of the PM.

### 2.3. Biomechanical Testing of the Specimens

After OS, the specimens were tested under a combination of external rotation and axial stress. A constant axial loading of 750 N was selected (average body weight). Additionally, 10 rotational cycles were performed, rotating the tibial plateau against the fixed foot, thus simulating an external rotation of the foot (see Figure 2c). Rotational load levels started at 2 Nm and were subsequently increased by 2 Nm up to 12 Nm.

For each load level, the maximal change in the distance between M1 and M2 was calculated.

### 2.4. Statistical Analsis

Possible differences in bone quality, measured via bone mineral density (BMD) obtained using DEXA scans, between left and right tibiae were analyzed using a t-test for paired samples. A general linear model for repeated measurements (rmGLM) was used, with the OS as between subject and the load levels of the repeated measures as the within factor. If one OS should be superior in terms of stability (marker movement), we expect a significant interaction effect. For the OS and the interaction effect additional to the *p*-value, the effect sizes are given as partial eta squared ($\eta^2_p$), owing to the small sample size. Values of 0.01, 0.06, or 0.14 indicate small, medium, or large effects, respectively [10]. Mean values

and 95% confidence intervals (CI) are used as statistical descriptive parameters and post hoc pairwise comparisons. For statistical analyses, SPSS version 26 (IBM SPSS Statistics for Windows, Armonk, NY, USA: IBM Corp.) was used. The significance level was set to $p = 0.05$.

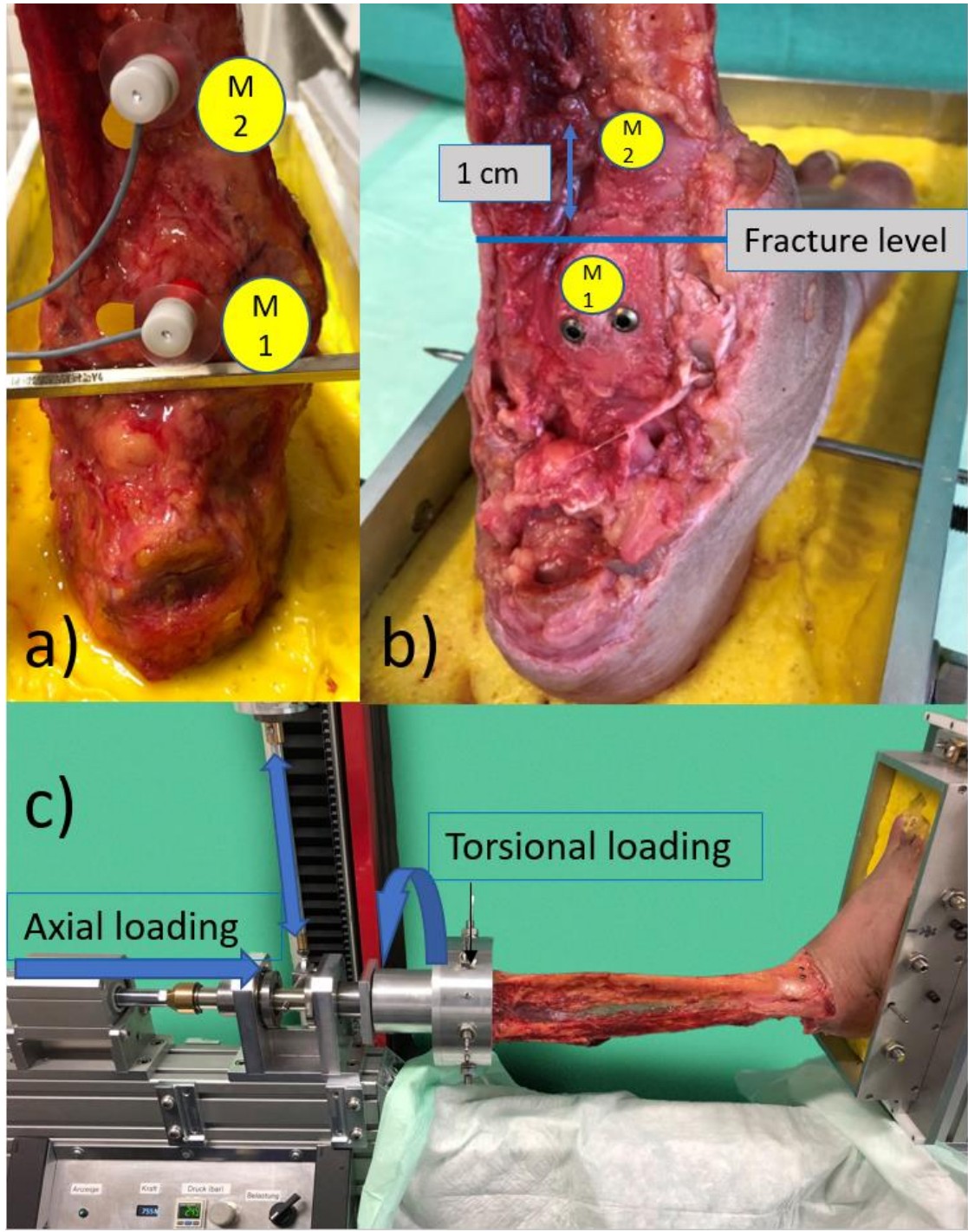

**Figure 2.** The figure shows in (**a**) the placed Zebris markers on the posterior malleolus (M1) and the distal tibia (M2), in (**b**) again the position of the markers M1 and M2 is shown, as well as the positioning of the M2 marker about 1 cm above the fracture level of the PM, and in (**c**) the test setup is shown with the lower leg clamped in the testing machine; the torsional loading was achieved by rotating the proximal tibia against the fixed foot.

## 3. Results

No significant difference in bone mineral density between the left and right side was found (left BMD: 0.82 ± 0.11 (mean and standard deviation: MW ± SD), right BMD: 0.87 ± 0.13, $p$ = 0.410).

For the primary stability, the regression model (rmGLM) showed no significance for OS, ($p$ = 0.378) at a medium effect size ($\eta^2_p$ = 0.065). The movement of the PM tended to be marginally greater for the OS with a.p. screws than the OS with p.a. screws. The interaction effect showed with $p$ = 0.691 and a small to medium ES ($\eta^2_p$ = 0.049), no significance, and, thus, no superiority of either OS. The trend was quasi parallel for both OS. The 95% confidence interval (Figure 3) illustrates for both sides that the movement increases as the load level increases ($p$ = 0.001). However, the lower and upper limits of the 95% confidence intervals of the two OS overlap at all load levels.

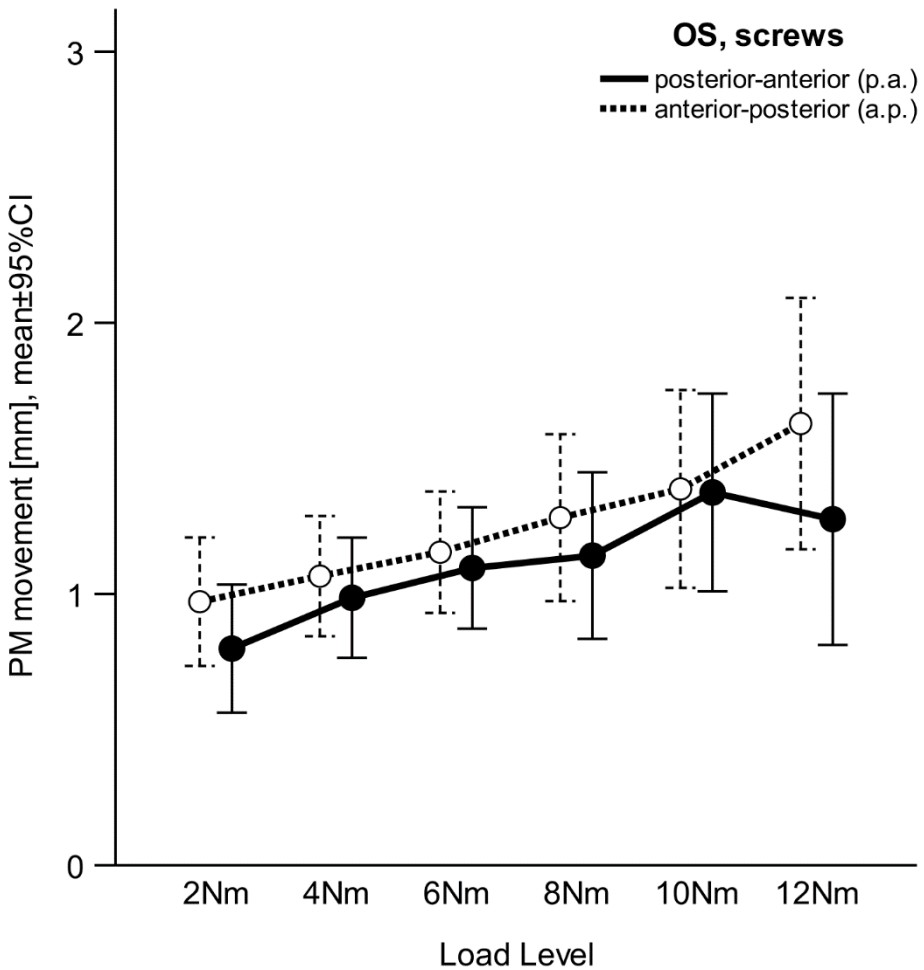

**Figure 3.** Movement of the posterior malleolus—Volkmann triangle—(PM) during load levels of 2 Nm to 12 Nm after osteosynthesis (OS) using posterior-anterior screws or anterior-posterior screws.

## 4. Discussion

In a recent study conducted by the authors of the present article [11], it was shown that an additive syndesmotic screw after a.p. fixation of the PM almost restores torsional stability compared to the uninjured initial state. Refixation of the PITFL alone, using screws placed from posterior, did not restore the initial stability of the upper ankle joint. It is not yet known whether the p.a or the a.p. stabilization leads to different primary stability after PM fixation. The aim of this biomechanical study was to investigate if the direction of screw osteosynthesis has an influence on the primary stability of the osteosynthesis. The results did not show a superiority of one OS regarding biomechanical primary stability.

Based on this result, the primary stability of the PM seemed quite comparable for both OS, with a marginal tendency of higher stability using p.a. screws. Thus, the results do not clearly correspond to the statement of Bartoníček et al., who report a biomechanical superiority of posterior osteosynthesis [5]. However, the biomechanical data from cadaver studies are only one of many criteria that should be considered when formulating general recommendations. The cadaver model only hints at the human joint without the muscle tonus.

A limitation of this biomechanical study is the small number of specimens. To address this, we used effect sizes to describe possible effects. Many other studies work with similarly small sample sizes [1,12–14]. The low numbers reflect a low availability of human specimens for test models.

The literature presents a discussion of the advantages and disadvantages of the dorsal approach and p.a. screw placement. Dorsal reduction and p.a. fixation seems to offer advantages for bone healing, adequate restoration of PITFL integrity, and restoration of tibiotalar articular surface anatomy [1,5]. Moreover, there are reports of better clinical postoperative outcome and less osteoarthritis [15,16]. On the other hand, some authors report higher complication rates such as injury to the fibular artery or flexor pollicis longus tendon for p.a. fixation [17,18].

The treatment with the a.p. screw is less invasive and can be performed faster compared to the more complex dorsal approach [19]. Based on our data showing comparable primary stability, stabilization can be achieved by a.p. screw fixation when reduction is easily achieved by ligamentotaxis. The p.a. treatment is the method of choice when reduction obstacles such as intermediate fragments are present.

We cannot make any statement about the maximum stability of the OS in patients with surgically treated ankle fractures, as we have not examined the OSed specimen until failure. Additional cyclic tests would be useful. In addition, it is difficult to transfer the in vivo load levels into ex vivo fracture models. The test setup attempts to reflect the movement with the highest lever arm on the fracture. The direction of screw insertion, p.a. or a.p., has no significant effect on the primary stability of osteosynthesis of the PM under torsional loading. If it is possible to achieve an adequate reduction result without repositioning obstacles, the a.p. technique could also be used. Due to the complication rates of the approach, the p.a. technique is reserved for fractures with inadequate reduction results due to ligamentotaxis. In the test model, we investigated the effect of screw direction on the primary stability of large fractures of the PM (type 4 according to Bartoníček and Rammelt) under torsional loading. In multifragmentary fractures, especially with intermediate fragments, the direction of the screws, or even the use of an alternative osteosynthesis technique (e.g., plate osteosynthesis), could offer biomechanical advantages. Further studies are necessary in this respect.

## 5. Conclusions

The direction of screw insertion, p.a. or a.p., has no significant effect on the primary stability of osteosynthesis of the PM under torsional loading. If it is possible to achieve an adequate reduction result without repositioning obstacles, the a.p. technique could also be used. Due to the complication rates of the approach, the p.a. technique is reserved for fractures with inadequate reduction results due to ligamentotaxis.

**Author Contributions:** Conceptualization, F.C.K., P.K., I.G. and J.H.; methodology, F.C.K., P.S. and J.H.; software, P.S. and S.D.; validation, F.C.K., P.K., P.S. and J.H.; formal analysis, F.C.K., P.S., P.K. and J.H.; investigation, F.C.K., P.K. and J.H.; resources, B.W., S.D., U.B. and G.O.H.; data curation, F.C.K., P.S., P.K. and J.H.; writing—original draft preparation, F.C.K.; writing—review and editing, P.S., P.K., G.O.H., U.B., B.W., S.D., I.G. and J.H.; visualization, F.C.K. and P.S.; supervision, G.O.H., S.D., U.B., B.W. and J.H.; project administration, F.C.K.; funding acquisition, B.W., G.O.H. and U.B. All authors have read and agreed to the published version of the manuscript.

**Funding:** This research received no external funding.

**Institutional Review Board Statement:** Approval was granted by the Ethics Committee of Jena University Hospital (approval no. 2020-1766-material).

**Informed Consent Statement:** Informed consent was obtained from all donors whose specimens were used in the study during their lifetime that their bodies could be used for educational and research purposes.

**Data Availability Statement:** The data presented in this study are available on request from the corresponding author.

**Acknowledgments:** We acknowledge support by the German Research Foundation and the Open Access Publication Fund of the Thueringer Universitaets- and Landesbibliothek Jena Project-Nr. 433052568.

**Conflicts of Interest:** The authors declare no conflict of interest.

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
