# Peer review of "How the Direction of Screws Affects the Primary Stability of a Posterior Malleolus Osteosynthesis under Torsional Loading: A Biomechanical Study"

_applsci, doi:10.3390/app12083833_

Round 1

Reviewer 1 Report

The authors may consider to add additional experiments/criterias for evluating if direction of screws affect the primary stability of PM, e.g., forces” in the comment section.
The reason is that I think additional data is helpful for strengthening the work’s conclusion that the direction of screws does not affect the primary stability of a posterior malleolus osteosynthesis under torsional force, though authors have already shown the result of PM movement under two different scenarios.

Author Response

Thank you very much for taking the time to read and review our manuscript. Please see the attachment for a detailed response to your review.

Reviewer 2 Report

The authors of this study undertake a small biomechanical test of the direction of screws in the context of primary stability of the posterior malleolus osteosynthesis. As this is a brief communication, the manuscript is necessarily concise. The small sample size of cadaver lower legs is a limitation, but the authors acknowledge it and I understand it is difficult to obtain larger sample size when working with cadaver material. Overall, I believe the findings are important and will be of use to clinicians and surgeons. My main concerns at the moment centre around the issues with text presentation and writing mechanics. In multiple places, the text requires polishing and revising to better convey the authors' meaning. Some of these are important for the delivery of the scientific content - e.g., in the Results section referring to BMD as DEXA is not appropriate- one is a variable, and the other is a technique. Because there are multiple of these instances in the manuscript, I have marked my comments on the attached PDF document of the manuscript for the authors' attention/perusal.

Author Response

(The authors gave the same response as above.)

Reviewer 3 Report

My first doubt: "torsion force" in the title. The force can be axial, lateral, .... Torque is related to torsion. The experiment should be described similarly to the "cake recipe" - I do not know how to repeat your experiment based on the work - what exactly was measured and how? The description doesn't have to be a long story. Without this information I cannot say anything about conclusions.

Author Response

(The authors gave the same response as above.)

Round 2

Reviewer 1 Report

The authors made great improvements for the manuscript revision (by adding new data/figures).  

Reviewer 3 Report

This version can be accepted.